# Augmented Deep Unrolling Networks for Snapshot Compressive Hyperspectral Imaging

## Abstract

Snapshot compressive hyperspectral imaging requires reconstructing a hyperspectral image from its snapshot measurement. This paper proposes an augmented deep unrolling neural network for solving such a challenging reconstruction problem. The proposed network is based on the unrolling of a proximal gradient descent algorithm with two innovative modules for gradient update and proximal mapping. The gradient update is modeled by a memory-assistant descent module motivated by the momentum-based acceleration heuristics. The proximal mapping is modeled by a sub-network with a cross-stage self-attention which effectively exploits inherent self-similarities of a hyperspectral image along the spectral axis, as well as enhancing the feature flow through the network. Moreover, a spectral geometry consistency loss is proposed to encourage the model to concentrate more on the geometric layer of spectral curves for better reconstruction. Extensive experiments on several datasets showed the performance advantage of our approach over the latest methods.

## 1 Introduction

Hyperspectral imaging captures a hyperspectral image (HSI) which is a 3D cube of intensities that represents the integrals the radiance of a real scene across a wide range of spectral bands. As an HSI provides rich spectral characteristics of objects of a scene, hyperspectral imaging has found wide applications in many areas, *e.g.*, agriculture, industry, and science. Snapshot compressive spectral imaging [1], often known as coded aperture snapshot spectral imaging (CASSI), is a compressed-sensing-based technique for rapid and efficient acquisition of HSIs. In contrast to traditional hyperspectral imaging techniques which use a sensor array for measuring the object at many spectral bands, the CASSI only collects a single coded 2D snapshot, which measures the object modulated by a physical mask and a disperser at the mixture of different wavelengths. A reconstruction algorithm is then called to reconstruct the 3D HSI from its 2D compressive snapshot.

Let $\boldsymbol{X} \in \mathbb{R}^{M \times N \times \Lambda}$ denote an HSI with spatial indices $m, n$ and spectral index $\lambda$. In general, the snapshot from a CASSI device can be expressed as the following [1]:

$$\boldsymbol{Y}(m, n) = \sum_{\lambda=1}^{\Lambda} \rho(\lambda)\varphi(m - J(\lambda), n)\boldsymbol{X}(m - J(\lambda), n, \lambda) + \boldsymbol{N}(m, n), \tag{1}$$

where $\rho(\cdot)$ is the spectral response of the camera, $\varphi(\cdot, \cdot)$ the coded aperture pattern, $J(\cdot)$ the dispersive function, and $\boldsymbol{N}$ the measurement noise. For convenience, we re-express it in a matrix-vector form:

$$\boldsymbol{y} = \boldsymbol{\Phi}\boldsymbol{x} + \boldsymbol{n}, \tag{2}$$

where $\boldsymbol{\Phi}$ denotes the measurement matrix determined by $\rho, \psi$, and $\boldsymbol{x}, \boldsymbol{y}, \boldsymbol{n}$ are the vectorized form of $\boldsymbol{X}, \boldsymbol{Y}, \boldsymbol{N}$, respectively. As Eq. (2) is an under-determined linear system with measurement noise, HSI reconstruction needs to solve an ill-posed inverse problem,

In recent years, deep learning has become a prominent approach for developing powerful solutions to HSI reconstruction; see *e.g.* [2, 36, 3–6, 3, 7–10].Most of them models the inverse mapping from the 2D snapshot to its corresponding HSI by a neural network (NN) trained over a dataset. Among existing designs of NN architecture, deep unrolling is the most popular one for HSI reconstruction, as it allows the inclusion of the physics of imaging. A typical deep unrolling network (DUN) unfolds an iterative scheme for solving some regularized variational model of (2), where the regularization-related parts are replaced by learnable NN modules. It can also be interpreted as a concatenation of the steps that alternates between an updating step and a refinement step: $\boldsymbol{x}^{(0)} \xrightarrow{Update} \boldsymbol{z}^{(0)} \xrightarrow{Refine} \boldsymbol{x}^{(1)} \xrightarrow{Update} \boldsymbol{z}^{(1)} \xrightarrow{Refine} \boldsymbol{x}^{(2)} \longrightarrow \cdots$. Despite extensive studies on HSI reconstruction, the practical need remains for the methods with better reconstruction accuracy.

The paper aims at developing a DUN for HSI reconstruction that brings noticeable performance improvement over existing deep NNs. The proposed DUN is based on the proximal gradient descent (PGD) algorithm [11, 12], one often seen iterative scheme for solving inverse problems in imaging. The PGD algorithm alternatively iterates between the following two steps:

1. A gradient descent step for updating the estimate of the image

2. A proximal mapping for refining the estimate via fitting some regularization term.

In comparison to existing DUNs for HSI reconstructions, there are three innovations in the design and training of the proposed one:

1. Updating step: Modeling the gradient descent step using an NN block with a momentum-motivated memory-assistant module which is implemented by long short-term memory.

2. Refinement step: Modeling the proximal mapping by a sub-NN with a across-stage self- attention module, for exploiting specific characteristics of HSIs and efficient feature flow.

3. Training loss: A spectral geometry consistency loss is proposed for regularizing the reconstruction with better accuracy.

**Learnable memory-assistant module**  In most existing DUNs for HSI reconstruction, the updating step usually is some pre-defined non-learnable process, *e.g.* gradient-based update. Gradient-based updates are in a zig-zag direction which slows down the movement to a minima. Also, the updates crawl near the minima or saddle points slowly as the gradient magnitude vanishes rapidly over there. A popular technique used for acceleration is the so-called *momentum* (*e.g.* RMSProp and Adam). Instead of using only the current gradient, momentum accumulates the gradients of the past steps to determine the direction to go, which helps move more quickly towards the minima as it dampens the zig-zag oscillations and builds the speed to quicken the convergence.

Motivated by the benefit brought by momentum in gradient-based update, we propose to learn an NN-based model for gradient-based update with the concept of momentum. As the effectiveness of momentum comes from its memory of the gradients of past steps, we propose an NN block with a memory-assistant mechanism such that it will leverage the gradient descents from previous stages, which is implemented using convolutional long short-term memory (ConvLSTM) units.

**Cross-stage self-attention module**  An HSI has its specific physical characteristics. One is the self-similarity and strong correlation along the spectral axis, as the entries along the spectral axis measure the same object region but at different wavelengths. To exploit such specific physical property of HSIs, we propose a self-attention module along the spectral axis. While self-attention is not completely new in image reconstruction, our implementation is different from existing ones by defining in a cross-stage manner.

One additional function for such a cross-stage self-attention module is to exploit the similarity of the features learned over different stages by forming a path between two different stages. Such similarities among the featured learned at different stages come from the fact that the role of refinement step is supposed to the same across different stages. The benefit of utilizing such similarity is two-fold. One is for more efficient feature delivery across the full stages, and the other is for enabling interactions among the features at different stages during the training.

**Loss on spectral geometry consistency**  In addition to the standard $\ell_1$ loss, a spectral geometry consistency loss is proposed for training the DUN for HSI reconstruction. Such a loss encourages the

model to concentrate more on the profile of spectral changes during reconstruction, which helps to improve the reconstruction accuracy as empirically observed.

## 2 Related Work

By imposing certain priors on HSIs, regularization is a widely-used approach to solving the problem of HSI reconstruction. The priors for natural images have been extended to HSIs, *e.g.*, sparsity prior in image gradients used in total variation [13, 14], sparsity prior under a learned dictionary [2, 15], and non-local self-similarity prior in the form of low-rankness for spatial-spectral patches [16–19]. These pre-defined priors are often insufficient for the HSIs with complex and diverse structures.

There is an increasing trend to use the implicit image prior encoded in a pre-trained or untrained NN for regularization. Plug-and-play methods [14, 20, 21] employ the NNs pre-trained on the denoising tasks of HSIs or natural images to regularize the reconstruction process. However, pre-trained denoising NNs are usually not very effective to handle the noise and artifacts generated in the iterative reconstruction process. Self-supervised learning methods [22, 23] use an untrained NN to re-parameterize the latent HSI and train it to match the observed snapshot. Such an online learning scheme is computationally expensive and cannot leverage the knowledge from external data.

It has been a prominent approach that to end-to-end train a DNN that maps a snapshot to the latent HSI; see *e.g.* [24, 5, 25, 26, 9, 8]. Many existing studies employ the DUN architecture*e.g.* [3, 6, 7, 4]. Recall that a DUN often consists of pairs of steps: one step for updating the estimate of the latent HSI and the other step for refining the estimate with a learnable prior. Most existing works focus on the latter, which can be viewed as a denoising NN that exploits different image priors, *e.g.*, spatial-spectral prior [3],non-local self-similarity prior [6], and patch-level Gaussian scale mixture prior [7].

**Learning updating steps in DUNs** Zhang *et al.* [4] replaced the operators $\mathbf{\Phi}, \mathbf{\Phi}^\top$ appearing in the gradient descent step of PGD by convolutions and residual blocks, with a channel attention block to estimate the step size in PGD from the estimate output by the previous stage. Different from that, we do not learn those operators but utilize them to have a better update step. Working on natural image recovery rather than on HSI reconstruction, Mou *et al.* [27] used a residual block to estimate the gradient descent step. In comparison, we use an LSTM to leverage the dependency between different stages for estimating the updating step.

**Self-attention for HSI reconstruction** Self-attention (SA) has been exploited in existing works for HSI reconstruction. Miao *et al.* [5] used a generative adversarial network with SA for the initial stage in the NN. Meng *et al.* [28] used three spatial-spectral SA modules to exploit the spatial-spectral correlation of an HSI. Hu *et al.* [9] develops a spatial-spectral attention module with efficient feature fusion. In comparison to these methods, ours treats spectral maps as tokens for SA and calculates the SA along the spectral dimension. This shares a similar idea with a parallel work [8] which also treats spectral maps as tokens in a transformer-based model. Different from it, we use SA in a cross-stage manner which enhances the feature flow at the same time.

**Training loss for HSI reconstruction** Most existing NNs for HSI reconstruction are trained by the standard mean-squared-error loss or $\ell_1$ loss. Hu *et al.* [9] introduced a frequency-domain loss to narrow the frequency-domain discrepancy between network predictions and ground truths. In comparison, the loss we proposed narrows the discrepancy in terms of spectral geometric changes.

## 3 Proposed Approach

The proposed DUN for HSI reconstruction is based on the PGD algorithm [11, 12] for the following optimization model regularized by the functional $\mathcal{R}$:

$$\min_{\boldsymbol{x}} \ \|\boldsymbol{y} - \mathbf{\Phi}\boldsymbol{x}\|_2^2 + \lambda\mathcal{R}(\boldsymbol{x}), \quad \lambda \in \mathbb{R}^+, \tag{3}$$

The PGD algorithm for solving Eq. (3) alternately iterates between two steps: gradient-descent (GD) step for updating the estimate, and proximal mapping (PM) step for refining the estimate by fitting the functional $\mathcal{R}$ with encoded image prior: For $k = 1, \cdots, K$,

$$[\text{GD}]: \quad \boldsymbol{u}^{(k)} = \boldsymbol{x}^{(k-1)} + \gamma^{(k)}\mathbf{\Phi}^\top(\boldsymbol{y} - \mathbf{\Phi}\boldsymbol{x}^{(k-1)}), \tag{4}$$

$$[\text{PM}]: \quad \boldsymbol{x}^{(k)} = \text{Prox}_{\mathcal{R}}(\boldsymbol{u}^{(k)}) \triangleq \operatorname*{argmin}_{\boldsymbol{x}'} \|\boldsymbol{x} - \boldsymbol{u}^{(k)}\|_2^2 + 2\gamma^{(k)}\mathcal{R}(\boldsymbol{u}^{(k)}). \tag{5}$$

130 where $\gamma^{(k)}$ denotes step size. Most existing DUNs focus on modeling the PM step (5) by an NN for a
131 data-driven prior. The GD step (4) usually is kept unchanged with the learnable parameter $\gamma^{(k)}$.

132 We propose a Memory-Assistant Descent (MAD) block to model the GD step (4) and a Cross-stage
133 Attentive Proximal (CAP) sub-network to model the PM step (5). The former functions as gradient
134 descent across different stages for momentum-motivated acceleration, which leads to a more efficient
135 update than that only using the gradient at current stage. The latter is to utilize the self-similarities
136 existing in an HSI with a cross-stage manner, which enable us to exploit special characteristics of
137 HSIs and fasten feature flow through the DNN. In short, the proposed NN, called MadcapNet, is the
138 concatenation of $K$ stages, each of which contains a pair of a MAD block and a CAP sub-network;
139 see Figure 1 for the diagram of MadcapNet.

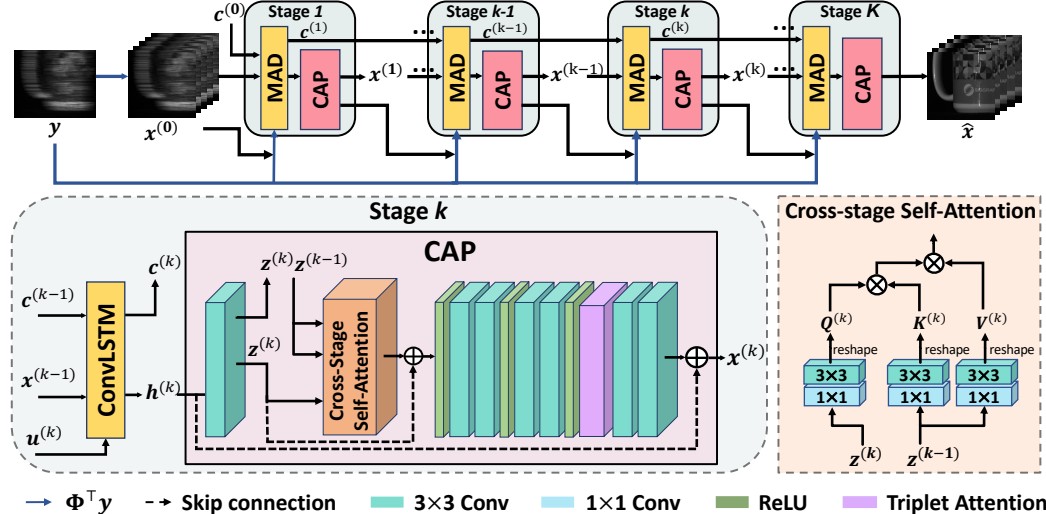

Figure 1: Diagram of the proposed augmented deep unrolling neural network for HSI reconstruction.

## 3.1 Memory-Assistant Descent Blocks

141 The MAD blocks are a set of ConvLSTM units [29] placed at each stage of the NN, which utilizes
142 the long-range dependencies among all cascading stages for momentum-assistant gradient update. In
143 each MAD block, the gradient map is defined by

$$\boldsymbol{u}^{(k)} = \boldsymbol{\Phi}^{\top}(\boldsymbol{y} - \boldsymbol{\Phi}\boldsymbol{x}^{(k-1)}) \tag{6}$$

144 is taken as the input for the $k$-th ConvLSTM unit, which introduces information on gradient descent.
145 Let $\boldsymbol{h}^{(k)}, \boldsymbol{c}^{(k)}$ denote the hidden state and cell state in the ConvLSTM at the $k$-th stage respectively,
146 where $\boldsymbol{h}^{(k)}$ is of the same size as $\boldsymbol{x}^{(k)}$. The MAD block is defined as

$$[\boldsymbol{h}^{(k)}, \boldsymbol{c}^{(k)}] = \text{ConvLSTM}(\boldsymbol{u}^{(k)}, \boldsymbol{x}^{(k-1)}, \boldsymbol{c}^{(k-1)}), \tag{7}$$

147 for $k = 1, \cdots, K$. Different from original ConvLSTM units which use the previous hidden state
148 $\boldsymbol{h}^{(k-1)}$ as input, we replace $\boldsymbol{h}^{(k-1)}$ by $\boldsymbol{x}^{(k-1)}$, the output from the CAP sub-network of the previous
149 stage. The motivation behind is to utilize the current gradient decent defined over $\boldsymbol{x}^{(k-1)}$. Then, $\boldsymbol{h}^{(k)}$
150 is used as the input of the CAP sub-network and $\boldsymbol{c}^{(k)}$ is fed to the MAD block at the next stage as an
151 accumulator of state information.

152 In the $k$-th stage, the ConvLSTM unit calculates $\boldsymbol{h}_k, \boldsymbol{c}_k$ by the following rules

$$\boldsymbol{c}^{(k)} = \boldsymbol{f}_k \odot \boldsymbol{c}^{(k-1)} + \boldsymbol{i}^{(k)} \odot \tanh(\boldsymbol{g}^{(k)}), \tag{8}$$

$$\boldsymbol{h}^{(k)} = \boldsymbol{o}^{(k)} \odot \tanh(\boldsymbol{c}^{(k)}), \tag{9}$$

where $\odot$ denotes Hadamard product, and $\boldsymbol{i}_k, \boldsymbol{f}_k, \boldsymbol{o}_k, \boldsymbol{g}_k$ denote the input gate, forget gate, output gate, and the intermediate result, respectively, which are calculated as follows:

$$\boldsymbol{i}^{(k)} = \text{sigmoid}(\boldsymbol{W}_{\text{mi}}\boldsymbol{u}^{(k)} + \boldsymbol{W}_{\text{xi}}\boldsymbol{x}^{(k-1)} + \boldsymbol{b}_{\text{i}}), \tag{10}$$

$$\boldsymbol{f}^{(k)} = \text{sigmoid}(\boldsymbol{W}_{\text{mf}}\boldsymbol{u}^{(k)} + \boldsymbol{W}_{\text{xf}}\boldsymbol{x}^{(k-1)} + \boldsymbol{b}_{\text{f}}), \tag{11}$$

$$\boldsymbol{g}^{(k)} = \boldsymbol{W}_{\text{mg}}\boldsymbol{u}^{(k)} + \boldsymbol{W}_{\text{xg}}\boldsymbol{x}^{(k-1)} + \boldsymbol{b}_{\text{g}}, \tag{12}$$

$$\boldsymbol{o}^{(k)} = \text{sigmoid}(\boldsymbol{W}_{\text{mo}}\boldsymbol{u}^{(k)} + \boldsymbol{W}_{\text{xo}}\boldsymbol{x}^{(k-1)} + \boldsymbol{b}_{\text{o}}), \tag{13}$$

where $\boldsymbol{W}_{**}$ are implemented by $3 \times 3$ convolutional layers with bias terms $\boldsymbol{b}_*$.

## 3.2 Cross-stage Attentive Proximal Sub-networks

The CAP blocks function as a learnable PM step (5) which refines the estimate from the MAD block. It can be understood as a denoising NN by interpreting the estimation residual as noise. Given $\boldsymbol{h}^{(k)}$ (of the same size as $\boldsymbol{x}$) from the MAD block as input, we map it to a feature tensor $\boldsymbol{z}^{(k)}$ via a convolutional layer, which is then processed by a cross-stage SA module. Afterward, the results are fed to a sequence of convolutional layers with rectified linear units (ReLUs) and a triplet attention [30]. The output with the same size as $\boldsymbol{x}$ is combined with the input $\boldsymbol{h}^{(k)}$ via a skip connection, yielding the reconstructed HSI $\boldsymbol{x}^{(k)}$ at the current stage. See Figure 1 for the details.

Recall that SA [31] relates input feature tokens to compute a refined feature representation. It first generates a key/query/value vector of length $d$ from each token, and all the key/query/value vectors are stored as $\boldsymbol{K}, \boldsymbol{Q}, \boldsymbol{V}$ respectively. Then, SA is calculated as follows:

$$\text{SA}(\boldsymbol{Q}, \boldsymbol{K}, \boldsymbol{V}) = \text{softmax}\left(\frac{1}{\sqrt{d}}\boldsymbol{Q}\boldsymbol{K}^\top\right)\boldsymbol{V}. \tag{14}$$

We treat each feature channel as a token so as to exploit the self-similarities among feature channels. Such tokens are aligned due to natural alignment of spectral slices of an HSI. In the $k$th stage, rather than use the feature $\boldsymbol{z}^{(k)}$ at current stage to calculate $\boldsymbol{K}^{(k)}, \boldsymbol{Q}^{(k)}, \boldsymbol{V}^{(k)}$, we only use $\boldsymbol{z}^{(k)}$ for $\boldsymbol{Q}^{(k)}$ while using the feature $\boldsymbol{z}^{(k-1)}$ of previous stage for $\boldsymbol{K}^{(k)}, \boldsymbol{V}^{(k)}$. Concretely, we calculate

$$\boldsymbol{Q}^{(k)} = \boldsymbol{W}_{\text{Qd}}^{(k)}\boldsymbol{W}_{\text{Qp}}^{(k)}\boldsymbol{z}^{(k)}, \boldsymbol{K}^{(k)} = \boldsymbol{W}_{\text{Kd}}^{(k)}\boldsymbol{W}_{\text{Kp}}^{(k)}\boldsymbol{z}^{(k-1)}, \boldsymbol{V}^{(k)} = \boldsymbol{W}_{\text{Vd}}^{(k)}\boldsymbol{W}_{\text{Vp}}^{(k)}\boldsymbol{z}^{(k-1)}, \tag{15}$$

where $\boldsymbol{W}_{(*p)}^{(k)}, \boldsymbol{W}_{(*d)}^{(k)}$ are $1 \times 1$ convolutions and $3 \times 3$ depth-wise convolutions respectively for better encoding spatial-channel context.

The motivation of the cross-stage strategy is as follows. The DUN architecture alternates between the update and the refinement. Since the CAP sub-networks at different stages play the same role of refinement, their extracted features should be highly correlated and the features extracted from the previous stage provide good initials for the corresponding ones at the next stage. However, the aforementioned pipeline does not utilize such correlations for more efficient training, which may result in a bottleneck for features flowing through the whole DUN. The proposed cross-stage SA scheme forms a path between two stages, which allows efficient feature transmission during inference and enhances feature interactions during training.

The multi-head strategy [31] is adopted for the cross-stage SA. First, we split the key/query/value matrices into $H$ heads along channel dimension: $\boldsymbol{Q}^{(k)} = [\boldsymbol{Q}_1^{(k)}, \cdots, \boldsymbol{Q}_H^{(k)}], \boldsymbol{K}^{(k)} = [\boldsymbol{K}_1^{(k)}, \cdots, \boldsymbol{K}_H^{(k)}]$, and $\boldsymbol{V}^{(k)} = [\boldsymbol{V}_1^{(k)}, \cdots, \boldsymbol{V}_H^{(k)}]$. Then, the output is calculated as

$$\boldsymbol{O}^{(k)} = \cup_{j=1}^{H}\text{SA}(\boldsymbol{Q}_j^{(k)}, \boldsymbol{K}_j^{(k)}\boldsymbol{V}_j^{(k)}), \tag{16}$$

which is reshaped for subsequent processing.

## 3.3 Loss function for Training

To better train a NN for HSI reconstruction, we propose an additional loss called spectral geometry consistency (SGC) loss. For an HSI $\boldsymbol{X} \in \mathbb{R}^{M \times N \times \Lambda}$, we define the geometry map $\mathcal{D}(\boldsymbol{x})$ as follows.

$$\mathcal{D}(\boldsymbol{X}) = \nabla_{\text{c}}(\text{sign}(\nabla_{\text{c}}\boldsymbol{X})) \in \{-1, 0, 1\}^{M \times N \times \Lambda}, \tag{17}$$

where $\nabla_c$ calculates the gradient along the spectral axis, and $\mathrm{sign}(\cdot)$ denotes element-wise sign function. For a spatial location $(m_0, n_0)$, $\mathcal{D}(\boldsymbol{X})[m_0, n_0, \cdot]$ indicates the wavelengths where the monotony of spectral values changes, which is one geometrical property of the spectral curve. Based on $\mathcal{D}$, the SGC loss emphasizes the geometrical layout consistency between the reconstructed HSI and ground truth.

Considering HSIs exhibit high spatial sparsity, the irrelevant dark regions are omitted for robustness. This is achieved by constructing a mask $\boldsymbol{M_X}$ that thresholds the max density along spectral dimension: $\boldsymbol{M_X}(m, n, \lambda) = 1$ if $\max_\lambda \boldsymbol{X}(m, n, \lambda) \geq \alpha$; and 0otherwise. Let $\boldsymbol{X}, \widehat{\boldsymbol{X}}$ denote the reconstructed HSI and its ground truth respectively. The SGC loss is defined as

$$\mathcal{L}_{\mathrm{sgc}} \triangleq \|\boldsymbol{M_X} \odot \mathcal{D}(\boldsymbol{X}) - \boldsymbol{M_{\widehat{X}}} \odot \mathcal{D}(\widehat{\boldsymbol{X}})\|_1. \tag{18}$$

By minimizing $\mathcal{L}_{\mathrm{sgc}}$, the HSI predicted by the NN is biased to the one with the same wavelength-density trends of ground truths, which helps to alleviate possible over-fitting. Then, the overall loss is

$$\mathcal{L} \triangleq \mathcal{L}_1 + \gamma \mathcal{L}_{\mathrm{sgc}} = \|\boldsymbol{X} - \widehat{\boldsymbol{X}}\|_1 + \gamma \|\boldsymbol{M_X} \odot \mathcal{D}(\boldsymbol{X}) - \boldsymbol{M_{\widehat{X}}} \odot \mathcal{D}(\widehat{\boldsymbol{X}})\|_1, \ \ \gamma \in \mathbb{R}^+. \tag{19}$$

## 4  Experiments

We implement MadcapNet with PyTorch. The stage number $K$ is set to 6. On all convolutional layers, the kernel sizes are all set to $3 \times 3$, and both the stride and padding number are set to 1. The head number $H$ for the self-attention in CAP blocks is set to 8. Regarding the training loss, we set $\alpha = \frac{5}{255}$ for $\boldsymbol{M_X}$ and $\gamma = 0.5$ for Eq. (19) The training is done via the Adam optimizer with a fixed learning rate of $10^{-4}$ and a maximal epoch number of 200. The same data augmentation scheme as [7] is adopted, including rotation and flipping. All the models are trained and tested on an NVIDIA GeForce RTX 1080Ti GPU. Our code will be released on GitHub. upon paper's acceptance. Following [7], Peak-Signal-to-Noise-Ratio (PSNR) and Structured SIMilarity (SSIM) index are adopted as the metrics to evaluate the reconstruction results quantitatively.

### 4.1  Evaluation on Synthetic Data

**CAVE and KAIST datasets**  Following [28, 7], we use the CAVE dataset [32] containing 32 HSIs with 31 spectral bands for training, and 10 scenes with 31 spectral bands from the KAIST dataset [14] for test. All these HSIs are cropped into patches with a spatial size of $256 \times 256$ and reduced to 28 wavelengths ranging from 450nm to 650nm via spectral interpolation. The snapshot measurements are generated by the $256 \times 256$ mask of CASSI used in [28].

Ten existing methods are chosen for comparison, including (a) two conventional methods: GAP-TV [13] and DeSCI [17]; (b) one self-supervised learning-based method: PnP-DIP [22]; and (c) seven supervised learning-based methods: $\lambda$-Net [5] HSSP [3], DNU [6], TSA-Net [28], DGSMP [7], HDNet [9], and MST-L [8]. The HSSP, DNU and DGSMP are based on DUNs. The HDNet and MST-L are from two latest works accepted in an upcoming conference.

The quantitative results are listed in Table 1, which are quoted from [8, 9] whenever possible and otherwise obtained with released codes. It can be seen that our approach significantly outperforms the compared ones. Specifically, MadcapNet shows remarkable superior performance over other DUNs. It also surpasses MST-L and HDNet (*i.e.* two latest methods) with an average PSNR gain of more than 1dB and 2dB respectively. Table 1 also compares the model complexity of different methods in terms of number of parameters and number of Giga Floating-point Operations Per Second (GFLOPS). Although our model contains ConvLSTM and self-attention blocks, it is still kept compact to maintain a relatively-low model complexity. Among all compared methods, our MadcapNet has the smallest number of GLOPS, and it is smaller than all other models except DNU. These results show the practicability of MadcapNet for real applications. To conclude, our approach can achieve the best trade-off between performance and model complexity.

**ICVL and Harvard datasets**  We also conduct experiments on the ICVL dataset [33] and the Harvard dataset [34], respectively. The ICVL dataset consists of 201 HSIs of real-world objects, each with 31 spectral bands collected from 400nm to 700 nm at a 10nm step. The Harvard dataset consists of 50 outdoor scenes, each with 31 spectral bands collected from 420nm to 720nm at a 10nm step.

Table 1: Quantitative results in PSNR(dB) (even rows) and SSIM (odd rows) on KAIST dataset.

| Method | #Param. | #GFLOPS | Scene#1 | #2 | #3 | #4 | #5 | #6 | #7 | #8 | #9 | #10 | Mean |
|---|---|---|---|---|---|---|---|---|---|---|---|---|---|
| GAP-TV | - | - | 26.82 | 22.89 | 26.31 | 30.65 | 23.64 | 21.85 | 23.76 | 21.98 | 22.63 | 23.10 | 24.36 |
| | | | 0.754 | 0.61 | 0.802 | 0.852 | 0.703 | 0.663 | 0.688 | 0.655 | 0.682 | 0.584 | 0.669 |
| DeSCI | - | - | 27.13 | 23.04 | 26.62 | 34.96 | 23.94 | 22.38 | 24.45 | 22.03 | 24.56 | 23.59 | 25.27 |
| | | | 0.748 | 0.62 | 0.818 | 0.897 | 0.706 | 0.683 | 0.743 | 0.673 | 0.732 | 0.587 | 0.721 |
| $\lambda$-net | 62.64M | 117.98 | 30.10 | 28.49 | 27.73 | 37.01 | 26.19 | 28.64 | 26.47 | 26.09 | 27.50 | 27.13 | 28.53 |
| | | | 0.849 | 0.805 | 0.870 | 0.934 | 0.817 | 0.853 | 0.806 | 0.831 | 0.826 | 0.816 | 0.841 |
| HSSP | - | - | 31.48 | 31.09 | 28.96 | 34.56 | 28.53 | 30.83 | 28.71 | 30.09 | 30.43 | 28.78 | 30.35 |
| | | | 0.858 | 0.842 | 0.823 | 0.902 | 0.808 | 0.877 | 0.824 | 0.881 | 0.868 | 0.842 | 0.852 |
| DNU | 1.19M | 163.48 | 31.72 | 31.13 | 29.99 | 35.34 | 29.03 | 30.87 | 28.99 | 30.13 | 31.03 | 29.14 | 30.74 |
| | | | 0.863 | 0.846 | 0.845 | 0.908 | 0.833 | 0.887 | 0.839 | 0.885 | 0.876 | 0.849 | 0.863 |
| PnP-DIP | 33.85M | 64.42 | 32.68 | 27.26 | 31.30 | 40.54 | 29.79 | 30.39 | 28.18 | 29.44 | 34.51 | 28.51 | 31.26 |
| | | | 0.890 | 0.833 | 0.914 | 0.962 | 0.900 | 0.877 | 0.913 | 0.874 | 0.927 | 0.851 | 0.894 |
| TSA-Net | 44.25M | 110.06 | 32.03 | 31.00 | 32.25 | 39.19 | 29.39 | 31.44 | 30.32 | 29.35 | 30.01 | 29.59 | 31.46 |
| | | | 0.892 | 0.858 | 0.915 | 0.953 | 0.884 | 0.908 | 0.878 | 0.888 | 0.890 | 0.874 | 0.894 |
| DGSMP | 3.76M | 646.65 | 33.26 | 32.09 | 33.06 | 40.54 | 28.86 | 33.08 | 30.74 | 31.55 | 31.66 | 31.44 | 32.63 |
| | | | 0.915 | 0.898 | 0.925 | 0.964 | 0.882 | 0.937 | 0.886 | 0.923 | 0.911 | 0.925 | 0.917 |
| HDNet | 2.35M | 154.00 | 34.95 | 32.52 | 34.52 | **43.00** | 32.49 | **35.96** | 29.18 | **34.00** | 34.56 | 32.22 | 34.34 |
| | | | 0.948 | 0.953 | 0.957 | **0.981** | 0.957 | 0.965 | 0.937 | 0.961 | 0.958 | **0.950** | 0.957 |
| MST-L | 2.03M | 28.15 | 35.40 | 35.87 | 36.51 | 42.27 | 32.77 | 34.80 | 33.66 | 32.67 | 35.39 | 32.50 | 35.18 |
| | | | 0.941 | 0.944 | 0.953 | 0.973 | 0.947 | 0.955 | 0.925 | 0.948 | 0.949 | 0.941 | 0.948 |
| MadcapNet | **1.51M** | **24.24** | **36.24** | **37.49** | **37.07** | 42.85 | **34.09** | 35.61 | **35.37** | 33.96 | **36.67** | **33.12** | **36.32** |
| | | | 0.951 | 0.961 | 0.963 | 0.981 | 0.962 | 0.966 | 0.949 | 0.962 | 0.960 | 0.948 | **0.961** |

Following the protocol of [3, 35], 50 HSIs in the ICVL dataset and 9 HSIs in the Harvard dataset are used for test respectively, and the rest samples for training. All HSIs for training and test are cropped into patches with a spatial size of $48 \times 48$, while keeping the band number unchanged. The snapshot measurements are generated by the $48 \times 48$ mask of CASSI used in [3].

Six existing methods are selected for comparison, including (a) a conventional method: SSNR [16]; and (b) six supervised learning-based methods: HSCNN [36],$\lambda$-Net [5], DNU [6], DTLP [37], and HDNet [9]. The DNU and DTLP use DUNs, and the HDNet is a latest method.

See Table 2 for the quantitative comparison. The results of the compared methods are cited from [37]. The proposed one outperformed all other methods, with more than $0.85$db PSNR improvement on both datasets. Such noticeable performance gains of MadcapNet over other DUNs again demonstrated the effectiveness of our network architecture.

Table 2: Quantitative results in PSNR(dB) and SSIM on ICVL and Harvard datasets.

| Dataset | Metric | SSNR | HSCNN | $\lambda$-Net | DNU | DTLP | HDNet | MadcapNet |
|---|---|---|---|---|---|---|---|---|
| ICVL | PSNR | 30.40 | 28.45 | 29.01 | 32.61 | 34.53 | 36.38 | **37.60** |
| | SSIM | 0.943 | 0.934 | 0.946 | 0.966 | 0.977 | 0.981 | **0.985** |
| Harvard | PSNR | 31.14 | 27.60 | 29.37 | 31.11 | 32.43 | 34.02 | **34.88** |
| | SSIM | 0.942 | 0.895 | 0.909 | 0.929 | 0.941 | 0.950 | **0.956** |

**Visual inspection** See Figure 2 for the visualization of HSI reconstruction results on two samples from the KAIST and Harvard datasets respectively. The spectral curves (density versus wavelength) correspond to the points marked by green boxes in the RGB references. In the legends of both figures, we provide the curve correlation value between the result of a compared method and the ground truth. Those values show that the HSIs reconstructed by the proposed MadcapNet have the highest correlation to the ground truths. We also visualize three spectral channels of an entire reconstructed HSI and zoom in the selected regions marked by yellow boxes. It can be seen that the results of MadcapNet are more visually pleasing than that of other compared methods, with a better reconstruction of structures.

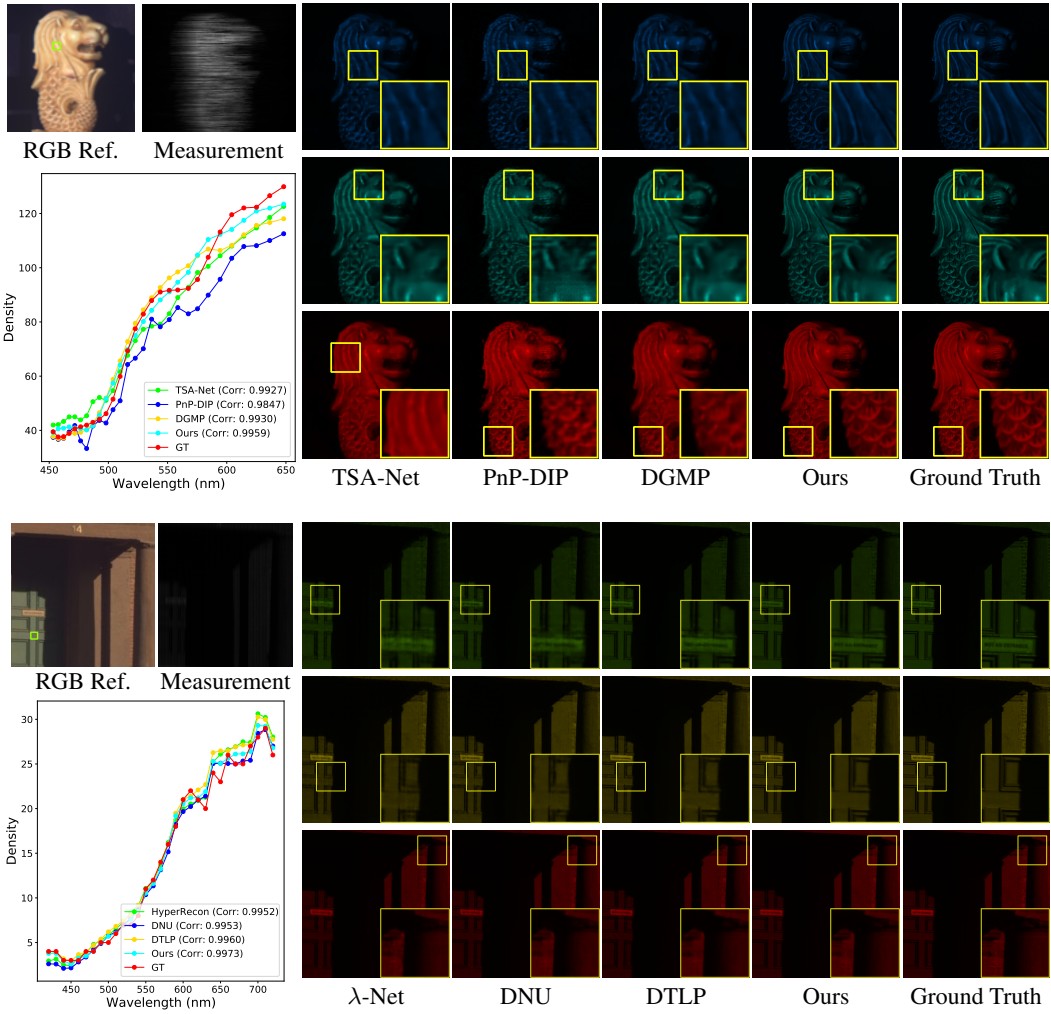

Figure 2: Visual comparison of HSI reconstruction on two samples from KAIST and Harvard datasets respectively. Left: spectra curves of the selected regions marked by green boxes. Right: reconstruction on the spectral channels.

## 4.2 Evaluation on Real Data

We also conduct a test on the real snapshots of spatial size $660 \times 714$ from [7, 28], which are captured by a real system with 28 wavelengths ranging from 450nm to 650nm and with 54-pixel dispersion in the column dimension. Following [7, 28], we use the mask associated with that real system to generate snapshots on both the CAVE and KAIST datasets, and then we inject 11-bit shot noise to the snapshots for simulating real situations. The resulting data is used to retrain our model. Due to the lack of ground truths in test data, we only compare the qualitative results of different methods. See Figure 3 for the reconstruction results on a real scene, and see more in the supplementary materials. The performance of MadcapNet is also good on the real data. This indeed has demonstrated the good generalization performance of our model.

## 4.3 Ablation Studies

Ablation studies are conducted on the KAIST dataset. We form some baselines by removing one or more main components of our approach. Concretely, we consider (a) replace the MAD blocks by the GD steps (4); (b) replace the cross-stage SA in the CAP network with the inner-stage SA which uses the features at current stage to calculate $\boldsymbol{K}^{(k)}, \boldsymbol{Q}^{(k)}, \boldsymbol{V}^{(k)}$ in (15); (c) replace the cross-stage SA

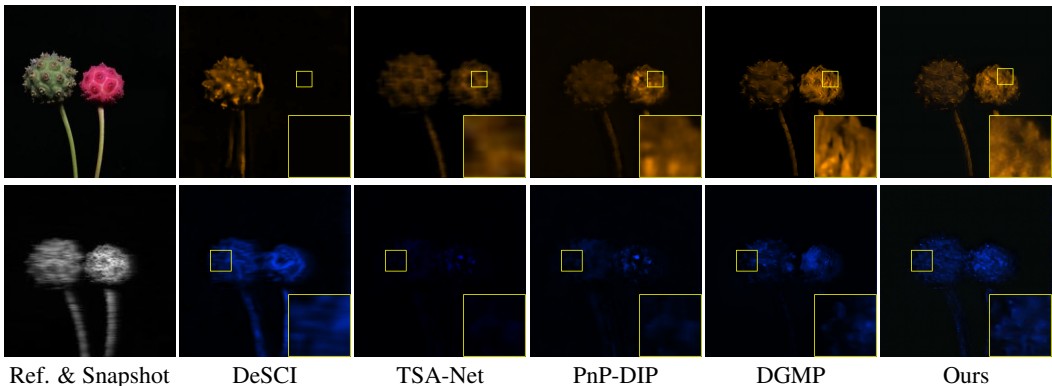

| Ref. & Snapshot | DeSCI | TSA-Net | PnP-DIP | DGMP | Ours |

Figure 3: Visual comparison of HSI reconstruction on real data, in terms of two spectral channels.

with a same number of convolutional layers; (d) remove the SGC loss $\mathcal{L}_{sgc}$. For a fair comparison, each baseline is configured to have (nearly) the same number of parameters as the original model, by uniformly increasing the channel numbers of convolutional layers. The results are listed in Table 3.

It can be seen that each main component in our approach plays an important role. Using the MAD blocks as an alternate to GD steps can improve PSNR by almost 1db. It also brings improvement across all baseline settings. Benefiting from the power of SA, the cross-stage SA brings noticeable PSNR gain. In addition, the SA utilized in the cross-stage manner leads to around 0.36dB improvement in PSNR over that utilized in the inner-stage manner. The SGC loss also has a solid contribution to the performance. See Figure 4 for an illustration of the effect of the SGC loss, where training with $\mathcal{L}_{sgc}$ makes the tendency of the predicted spectral curves closer to ground truths. See also supplementary materials for more results.

Table 3: Results in ablation studies on KAIST dataset.

| Metric | w/o MAD | w/o CAP | Cross→Inner | w/o $\mathcal{L}_{sgc}$ | Original |
| --- | --- | --- | --- | --- | --- |
| PSNR(dB) | 35.35 | 35.80 | 35.96 | 35.53 | **36.32** |
| SSIM | 0.947 | 0.956 | 0.958 | 0.951 | **0.961** |

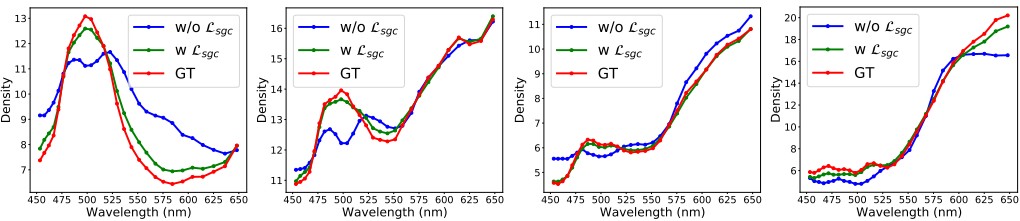

Figure 4: Spectra of selected regions on Scene#1 (first two) and Scene#5 (last two) of KAIST dataset.

## 5 Conclusion

In this paper, we proposed an augmented DUN for CASSI-based hyperspectral imaging. The proposed DUN is based on the unfolding of PGD, with three-fold augmentations: momentum-motivated ConvLSTM-assistant module for improving the gradient descent steps, a sub-network with cross-stage self-attention for exploiting self-similarities of an HSI and enhancing feature flow simultaneously, and a loss to induce predictions biased to spectral geometry consistency. The combination of these augmentations leads to noticeable performance improvement in HSI reconstruction, which were demonstrated by extensive experiments. The proposed DUN also sees its potential application to other compressive imaging problems. We will study it in the future.

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
