# Augmented Deep Unrolling Networks for Snapshot Compressive Hyperspectral Imaging (Supplementary Materials)

**Erratum** There is a mistake in Figure 2 of our main manuscript, where another region on the result of TSA-Net is wrongly shown. Please refer to Figure 1 for the correct version. It can be seen that our approach yielded better visual quality, in comparison to TSA-Net.

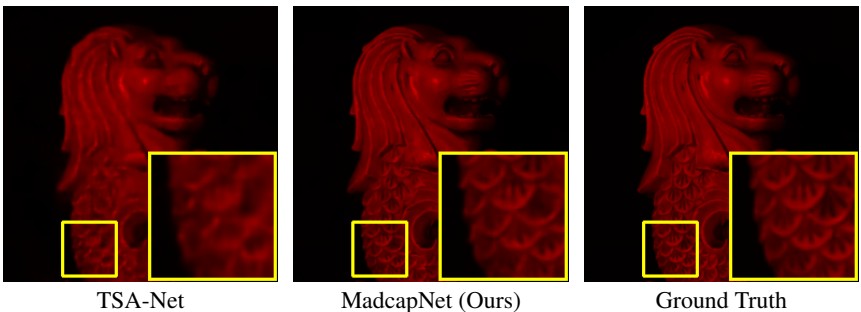

| TSA-Net | MadcapNet (Ours) | Ground Truth |

Figure 1: Visual comparison of HSI reconstruction on Figure 2 in main manuscript.

## 1 An Additional Ablation Study on Memory-Assistant Descend Blocks

An additional ablation study is conducted on the KAIST dataset to evaluate the benefit of using memory units in the updating step, where we replace the MAD blocks with convolutional layers. For a fair comparison, we use 3 convolutional layers of 56 channels to replace each MAD block so that the number of parameters is kept nearly the same as the original one, and concatenate $u^{(k)}$ and $x^{(k-1)}$ as the input. The PSNR(dB)/SSIM result of this new baseline is $35.48/0.949$, while the original one is $36.32/0.961$. In other words, the introduction of memory units in our approach does bring noticeable performance improvement over a plain implementation without memory units.

## 2 More Examples for Studying the Impacts of SGC Loss

In addition to Figure 4 in the main manuscript, Figure 2 here provides more examples for demonstrating the effectiveness of the SGC loss.

## 3 More Examples of Reconstructed HSIs for Visual Comparison

Please see Figures 4,5,6,7 for more examples of restructured HSIs for visual comparison.

Submitted to 36th Conference on Neural Information Processing Systems (NeurIPS 2022). Do not distribute.

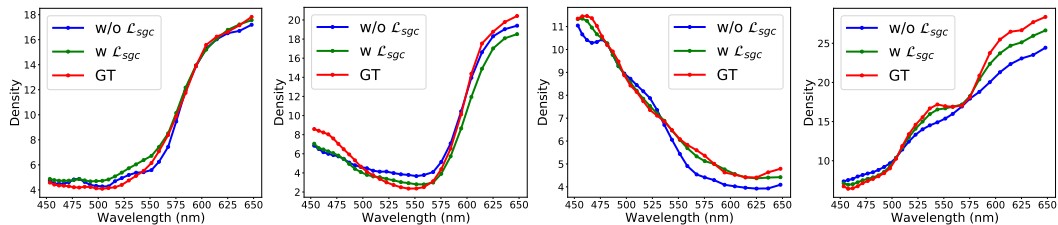

Figure 2: Spectra of selected regions in Scene #2 (left two), #4 and #10 of KAIST dataset.

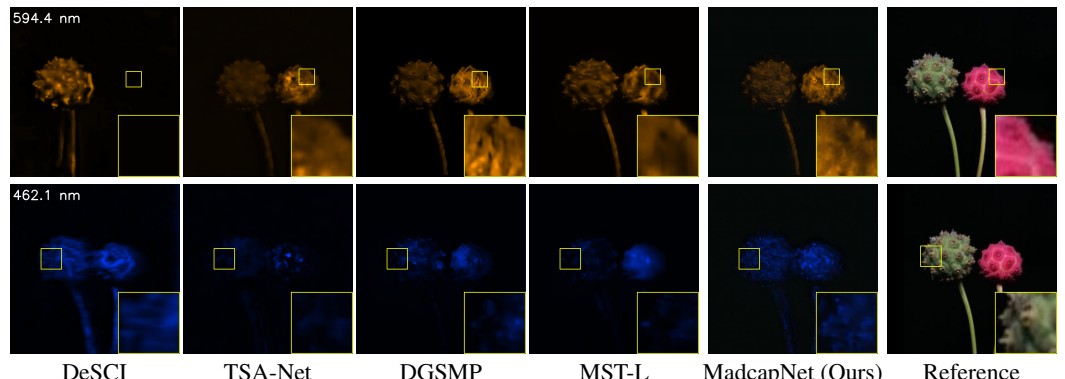

Figure 3: Visual comparison of HSI reconstruction on real data, in terms of two spectral channels.

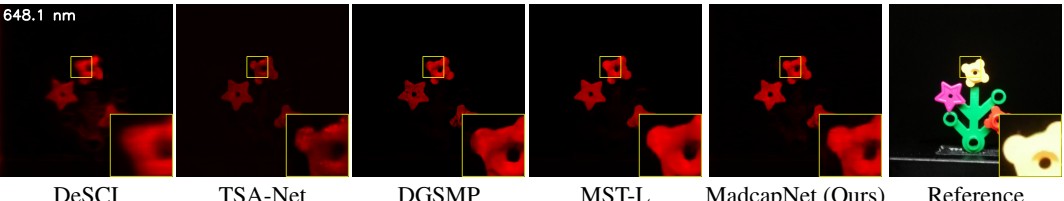

Figure 4: Visual comparison of HSI reconstruction on real data in one spectral channel.

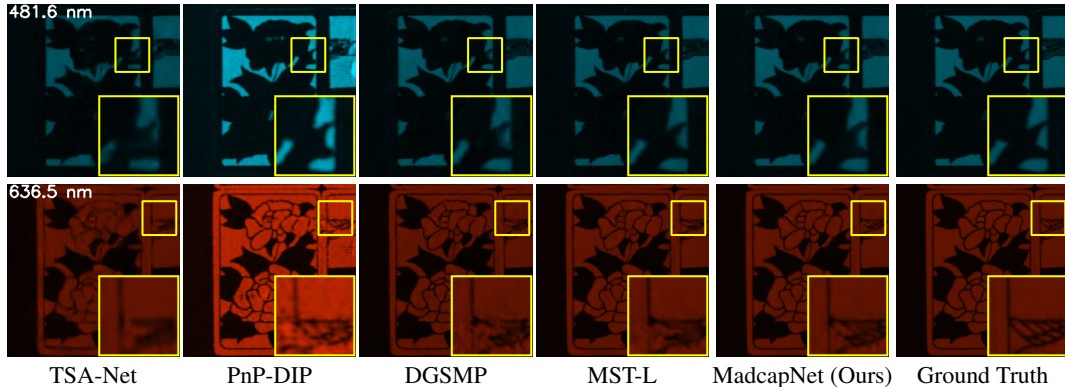

Figure 5: Visual comparison of reconstruction results on a sample from KAIST dataset, in terms of two spectral channels.

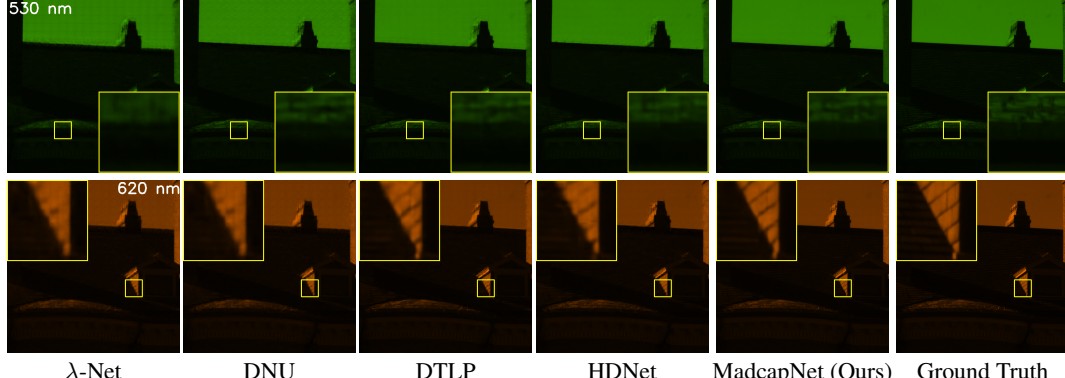

λ-Net          DNU          DTLP          HDNet          MadcapNet (Ours)   Ground Truth

Figure 6: Visual comparison of reconstruction results on a sample from Harvard dataset, in terms of two spectral channels.

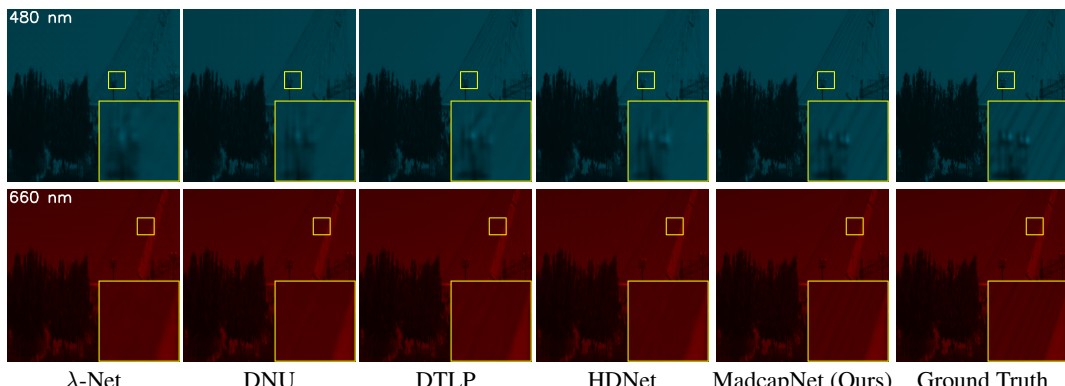

λ-Net          DNU          DTLP          HDNet          MadcapNet (Ours)   Ground Truth

Figure 7: Visual comparison of reconstruction results on a sample from ICVL dataset, in terms of two spectral channels.

## 4 Boarder Impacts and Limitations

The techniques proposed in our enhanced deep unrolling networks for snapshot compressive hyper-spectral imaging, such as the memory-assistant descend module and the cross-stage self-attention module, may also see their potential applications in the deep unrolling networks used for other image reconstruction or recovery problems. In addition, the proposed spectral geometry consistency loss function may inspire other loss functions for various signal processing tasks which consider the global geometric properties of the target signals being reconstructed. Our work serves as a low-level image reconstruction module, and thus it does not have a direct social impact. However, the technologies and applications developed based on our work can be either socially beneficial or harmful.

Similar to existing approaches, the proposed one is based on supervised learning which requires ground truth hyperspectral images for training. When the ground truths are of a limited number due to the cost or challenges of data acquisition, the performance of our model may decrease much. How to effectively train the model in data-limited scenarios will be our future work.