# OpenReview forum: "Augmented Deep Unrolling Networks for Snapshot Compressive Hyperspectral Imaging"
_NeurIPS.cc/2022/Conference — NeurIPS 2022 Submitted_

### Official Review · Reviewer_fWQz · 2022-07-11

**Rating:** 4
**Confidence:** 5
**Soundness:** 2 fair
**Presentation:** 2 fair
**Contribution:** 2 fair

**Summary:**

This paper proposes a proximal gradient descent (PGD) based deep unrolling network for snapshot compressive hyperspectral imaging, which employs a memory assistant descent module for gradient descent and a sub-network with cross-stage self-attention for proximal mapping. The experimental results show that the proposed method outperforms the compared methods.

**Questions:**

Please see strength and weakness

**Limitations:**

Yes

**Strengths And Weaknesses:**

Strength
+ Deep unrolling network is an important technique for snapshot compressive sensing.
+ Modeling correlation between different iteration stages in deep unrolling network is a good idea to improve the performance.
Weakness
- Novelty. Memory assistant deep unrolling network has been proposed in previous work, such as [1]. The relationship and the difference should be discussed. Besides, the comparison between these two methods is also necessary.
- Interpretability. Deep unrolling network is famous for its interpretability, compared with conventional deep network. However, adding the connection between different iteration stages weaken the interpretability. Moreover, in this paper, employing ConvLSTM to model the correlation between gradient descent in different iteration stages, which does not follow the PGD iteration. Besides, for hyperspectral snapshot compressive sensing, employing other optimization methods, e.g, ADMM, always have a fast close-form solver for data-fidelity. Thus, gradient descent and further memory-assistant module may be unnecessary.
- Experiment. PSNR and SSIM only evaluate the spatial fidelity between reconstructed HSI and groudtruth. Spectral fidelity evaluation metrics, e.g., SAM and ERGAS, are also necessary, especially the proposed method proposes a spectral loss.
- Writing. The citation of RMSProp and Adam is lacking in line 61. There lacks blank between 0 and otherwise in line 195. Please checking the typos of this manuscript.

---

> ### Author Response · Authors · 2022-08-02
> **Response to the comments from Reviewer fWQz**
>
> We sincerely thank the reviewer for his/her constructive comments. Please see below for our point-wise response.
>
> ***1、 Memory assistant deep unrolling network has been proposed in previous work, such as [1]. The relationship and the difference should be discussed. Besides, the comparison between these two methods is also necessary.***
>
> We cannot find the exact reference given by the reviewer. The closest work we found on memory-assistant deep unrolling networks (DUNs) for image reconstruction is the following one:
>
> ''Memory-augmented deep unfolding network for compressive sensing, ACM MM 2021.''
>
> The DUN proposed in this paper incorporates memory blocks into the module for proximal mapping, while ours uses memory blocks for the gradient descend module and uses cross-stage self-attention for the proximal mapping modules. This method is proposed for compressive sensing of natural images, not HSI. We adopt it for our problem and set its stage number to 13 so as to keep its model size close to ours. Its mean PSNR(dB) result on KAIST dataset is 34.78dB which is lower than ours (36.32dB) and even lower than ours trained without SGC Loss (35.53db).
>
> ***2、Interpretability. Deep unrolling network is famous for its interpretability, compared with conventional deep network. However, adding the connection between different iteration stages weaken the interpretability. Moreover, in this paper, employing ConvLSTM to model the correlation between gradient descent in different iteration stages, which does not follow the PGD iteration. Besides, for hyperspectral snapshot compressive sensing, employing other optimization methods, e.g, ADMM, always have a fast close-form solver for data-fidelity. Thus, gradient descent and further memory-assistant module may be unnecessary.***
>
> We agree that better interpretability is one property of DUNs. Depending on which iteration scheme it is built upon, different DUNs have different implementations. While ADMM-unrolled networks have a closed-form solver for data-fidelity, it does not necessarily perform better gradient-descent-based ones; see e.g.
>
> ''Dynamic proximal unrolling network for compressive imaging. arXiv, 2021'''
>
> The proposed method is about unrolling a gradient descent-based method with a momentum-based acceleration. The momentum-based acceleration is a widely-used technique; See e.g. the following reference for the details:
>
> ''On accelerated proximal gradient methods for convex-concave optimization. SIAM J Optimization, 2008.''
> In our opinion, the interpretation of deep unrolling network comes from that the regularization step is replaced by a learnable NN module. That is, the NN sub-module is some neutralized regularization step. Then, replacing the updating step related to the fidelity term with a learnable NN module can have the same argument, That is, such an NN sub-module is some neutralized updating step for more effective updating. It does not severely impact the interpretability of unrolling NN. Indeed, the recent studies on unrolling NNs did focus on replacing gradient descent steps with learnable modules (e.g. [5,28]) or connecting consequent stages (e.g. the reference mentioned in the previous response). The experiments showed that the introduced learnable module for the gradient updating step indeed led to noticeable performance improvement.
>
> ***3、 Experiment. PSNR and SSIM only evaluate the spatial fidelity between reconstructed HSI and ground-truth. Spectral fidelity evaluation metrics, e.g., SAM and ERGAS, are also necessary, especially the proposed method proposes a spectral loss.***
>
> Thanks for the suggestion. We calculated these two metrics, i.e., SAM (Spectral Angle Mapping) and ERGAS (Relative Dimensionless Global Error in Synthesis), which are shown in the following table. It can be seen that our method also outperformed other compared methods in terms of these two metrics. We will add these results in revision.
>
> SAM/ERGAS results of different methods.
> | Dataset | $\lambda$-Net  | TSA-Net    | DGSMP      | HDNet      | MST        | MadcapNet  |
> |---------|--------------|------------|------------|------------|------------|------------|
> | KAIST   | 19.71/108.63 | 8.75/90.78 | 8.94/31.50 | 6.68/25.64 | 7.47/27.72 | **5.42**/**10.07** |
>
> | Dataset | SSNR       |    HSCNN    | λ-Net      | DNU        | DTLP       | HDNet      | MadcapNet  |
> |---------|------------|-------------|------------|------------|------------|------------|------------|
> | ICVL    | 1.83/50.60 | 1.83/61.57  | 2.52/50.60 | 2.21/37.53 | 1.72/30.07 | 1.07/9.37  | **0.694**/**5.54** |
> | Harvard | 4.58/74.91 | 6.24/105.89 | 7.62/62.51 | 5.78/73.53 | 5.16/62.51 | 4.38/48.39 | **2.40**/**24.63** |
>
> ***4、 Writing. The citation of RMSProp and Adam is lacking in line 61. There lacks blank between 0 and otherwise in line 195. Please checking the typos of this manuscript.***
>
> Thanks for the careful reading. We will correct the typos in revision.

---

### Official Review · Reviewer_y9Ct · 2022-07-11

**Rating:** 6
**Confidence:** 3
**Soundness:** 3 good
**Presentation:** 2 fair
**Contribution:** 3 good

**Summary:**

The paper proposes a new unrolled neural network architecture for the problem of snapshot compressive hyperspectral imaging.
The architecture is inspired from proximal gradient descent, where both the gradient update and the proximal mapping steps are replaced by two recurrent modules.
The former (Memory-Assistant Descent) is based on ConvLSTM, and the latter (Cross-stage Attentive Proximal sub-network) uses a combination of convolutions, self-attention and triplet attention.
The authors compare against multiple recent approaches on several datasets.


**Questions:**

* Regarding generalization to real data:
  * As previously stated, visual comparison on real data is not provided against the most performant methods (HDNet, MST-L). This would help assessing the performance of the method.
  * A convincing argument would be to provide quantitative results on synthetic data **with noisy measurements**, similar to table 2 of the paper TSA-Net, where the authors argue that shot noise is realistic in this setting.
* There is no evidence that the unrolled architecture is beneficial with this method. An ablation study with a few different number of iterations/stages would be more convincing (including the limit case of a single iteration).
* Since both the Gradient Update and the Proximal Mapping modules benefits from using information across several stages/iterations, what is the justification for using different recurrent architectures (ConvLSTM and "Cross-stage self-attention") for tasks which are so similar in spirit ? If the authors disagree with this last statement, please explain why.
* The use of triplet attention for the CAP module is mentioned, but it would be helpful to develop a bit more why it is used here.
* It should be made clearer that the results presented in table 1 and table 2 are for noise-free synthetic data.
* What is the batch size used ?
* Minor:
  * The same notation is used to denote convolutions (eq. 10-13 and 15) and matrix multiplication (eq. 14). It would be clearer to denote the convolution operation with a dedicated symbol (see for example the ConvLSTM paper), and more coherent as the Hadamard operator has its own explicit symbol.
  * In eq 16, the symbol used for concatenation is ambiguous.
  * l. 202: it is unclear what the authors refer to as "padding number". Another formulation would be clearer.
  * l. 79: supposed to **be** (I guess)
  * eq. 10-11: missing term with **C^(k-1)**
  * l. 201: please provide citation for PyTorch
  * Table 1:
    * the table is difficult to read. It would benefit from enhancing the separation between methods (spacing or horizontal lines).
    * the number of parameters of DNU should be in bold
  * fig. 2-3 of main paper and fig. 3 of supplemental: DGMP -> DGMSP ?
  * on visual comparison, it would be helpful to know which spectral channel are shown.
  * on visual comparison on real data, it would be helpful to display the zoomed region on the reference and snapshot, as it is done with the results
  * l. 149: decent -> descent



**Limitations:**

* The broader impact section is not included in the 9 pages limit. Instead it is addressed in the supplemental material.
* The authors do not address the fact that their quantitative experiments are only valid on noise-free measurements only.
* The authors adequately mention the limitations of the supervised framework, i.e. its dependency on the amount of data available.

**Strengths And Weaknesses:**

# Strengths
* The method achieves very good performances on noise-free data
* The performances are compared against several recent methods
* Several datasets are used to show the superior performances of the proposed method
* Ablation studies are provided for multiple parts of the method, allowing to validate the overall significance of the main contributions

# Weaknesses
* All quantitative results shown are valid on noise-free measurements only.
  * No quantitative results are provided for noisy measurements, although the image formation model described in the introduction explicitly mentions the presence of noise on measurements.
  * This is even more dissonant as the model is fine-tuned on synthetic **noisy** data (shot noise) before applying it to real data for visual assessment, which indicates that the noise-free setting is not very realistic.
* The authors claim that the visual results on real data is sufficient evidence of the good generalization of the method on real measurement data (i.e. with noise). This is not a convincing argument as it is very difficult to see any improvement of the proposed method over its competitors on real data.
* All quantitative results are provided without error bars.
* No visual comparison is provided against the most competitive methods (HDNet, MST-L) on real data.

---

> ### Author Response · Authors · 2022-08-02
> **Response to the comments from Reviewer y9Ct (Cont.)**
>
> ***5、There is no evidence that the unrolled architecture is beneficial with this method. An ablation study with a few different number of iterations/stages would be more convincing (including the limit case of a single iteration)***
>
> As suggested, we reduce the stage number to $K=3$ and $K=2$ respectively to form two baselines for an ablation study. For a fair comparison, we double (for $K=3$) or treble (for $K=2$) the number of convolutional layers in the CAP module, which makes the model size of each baseline similar to that of the original model. The PSNR results of the baselines on the KAIST dataset are 33.67dB for $K=3$ and 31.86dB for $K=2$, while that of the original model ($K=6$) is 36.31dB. This suggests that the unrolled architecture is beneficial to our method. Note that for the limit case of a single iteration, the Conv-LSTM and cross-stage SA is not applicable, and thus we do not evaluate it.
>
> ***6、Since both the Gradient Update and the Proximal Mapping modules benefit from using information across several stages/iterations, what is the justification for using different recurrent architectures (ConvLSTM and "Cross-stage self-attention") for tasks which are so similar in spirit? If the authors disagree with this last statement, please explain why.***
>
> The Conv-LSTM is used for the gradient update so as to better model the temporal dependency of the gradient vectors among different iterations. Using cross-stage connections like what cross-stage self-attention does is not as efficient as LSTM for capturing the long-term dependencies.
> The cross-stage self-attention used for the proximal mapping as self-attention effectively exploits the high redundancy among different features of an HSI. The Conv-LSTM is not used here as LSTM is not as effective as self-attention in this aspect.
>
> ***7、The use of triplet attention for the CAP module is mentioned, but it would be helpful to develop a bit more why it is used here.***
>
> Attention mechanisms are widely used in deep NNs for improving the adaptivity to test samples. We implement the attention in our NN by the triplet attention module, as it is a light-weight attention module that inter-dimensional dependencies by simple rotation operations and encodes inter-channel and spatial information with negligible computational overhead. Using triplet attention allows for better exploiting the spatial-spectral correlation of HSIs for reconstruction. Removing the triplet attention will lead to around 0.13/0.15/0.12dB PSNR decrease on the KAIST/ICVL/Harvard dataset. We will add the discussion in revision.
>
> ***8、What is the batch size used?***
>
> The batch size is set to 4.
>
> ***9、Regarding the minor points***:
>
> Thanks for the careful reading. We will fix those minor issues, and refine the notations and format as suggested. Regarding point (e), $c^{\(k-1\)}$ is the cell state of LSTM which does need to be used in Eq. (10)-(11).

---

> ### Author Response · Authors · 2022-08-02
> **Response to the comments from Reviewer y9Ct**
>
> ***1、''All quantitative results shown are valid on noise-free measurements only. No quantitative results are provided for noisy measurements, although the image formation model described in the introduction explicitly mentions the presence of noise on measurements. This is even more dissonant as the model is fine-tuned on synthetic noisy data (shot noise) before applying it to real data for visual assessment, which indicates that the noise-free setting is not very realistic. A convincing argument would be to provide quantitative results on synthetic data with noisy measurements, similar to table 2 of the paper TSA-Net, where the authors argue that shot noise is realistic in this setting.''***
>
> ***'' It should be made clearer that the results presented in Table 1 and Table 2 are for noise-free synthetic data.''***
>
> ***'' The authors do not address the fact that their quantitative experiments are only valid on noise-free measurements only.''***
>
> Our quantitative evaluations mainly follow the widely-used settings used in existing studies (e.g., [7-10,27,30]) which use noise-free synthetic measurement data. To address the comments above, we conduct an additional experiment which follows a setting used in the paper of TSA-Net. The measurements in training data are corrupted with 11-bit shot-noise, while the ones in test data are corrupted with 11-bit,12-bit and 13-bit shot-noise respectively. All the compared models are retrained on the noisy measurements. See Table A in the following for the results, where our method still performs the best among the compared methods.
>
> Table A: PSNR(dB) results (mean$\pm$std.) on KAIST dataset with shot noise.
>
> | Noise  | $\lambda$-Net      | TSA-Net    | DGSMP      | HDNet      | MST-L        | Ours       |
> |--------|------------|------------|------------|------------|------------|------------|
> | 10-bit | 27.01$\pm$0.07 | 28.02$\pm$0.03 | 29.30$\pm$0.08 | 31.09$\pm$0.05 | 31.09$\pm$0.04 | 32.23$\pm$0.04 |
> | 11-bit | 27.36$\pm$0.05 | 28.34$\pm$0.03 | 29.89$\pm$0.07 | 31.09$\pm$0.03 | 31.53$\pm$0.03 | 32.80$\pm$0.03 |
> | 12-bit | 27.56$\pm$0.06 | 28.58$\pm$0.02 | 29.16$\pm$0.05 | 31.21$\pm$0.03 | 31.77$\pm$0.03 | 33.02$\pm$0.02 |
>
> We will make it clearer that the quantitative evaluations in Table 1 and 2 are based on noise-free measurements. In addition, we will add the evaluation of noisy measurements above to the paper.
>
> ***2、 The authors claim that the visual results on real data is sufficient evidence of the good generalization of the method on real measurement data (i.e. with noise). This is not a convincing argument as it is very difficult to see any improvement of the proposed method over its competitors on real data.***
>
> It is indeed not easy to see the difference among the visual results in the figures, as no ground truth to compare and the existence of dark background. If we leverage the RGB references and zoom in, we can find that our reconstruction results are less blurry than that of PnP-DIP, contain more details than that of DeSCI and TSA-Net, and have more similar structures to the RGB references than DGMP. We will improve the presentation with a better visualization.
>
> ***3、All quantitative results are provided without error bars.***
>
> Since the experimental settings we follow use fixed training sets and fixed test sets, the results from multiple-time training differ very little. As a result, existing studies do not report error bars for the quantitative results. To address this comment, we list the standard deviation (std.) of the PSNR on the noisy measurements in Table A in the response to the 1st comment. The randomness comes from the measurement noise and thus the standard deviation can be used to further check the robustness of the models. It can be seen that the standard deviations of all compared models are sufficiently small, which demonstrated their robustness.
>
> ***4、 Visual comparison on real data is not provided against the most performant methods (HDNet, MST-L). This would help assessing the performance of the method.***
>
> Both HDNet and MST-L are published in CVPR 2022. Their arVix versions are available right before the submission deadline of NeurIPS, and their codes and full reconstruction results were published after the submission deadline of NeurIPS. Thus, we could not include them for qualitative comparison in the manuscript. Even that, we had quoted their quantitative results for comparison. In the supplementary materials updated during rebuttal, we have included the visual results of MST-L, the second-best performer on the KAIST dataset for comparison.
> For the ICVL and Harvard datasets where MST-L is not compared quantitatively, we include the visual results of HDNet for comparison. Please see the updated supplementary materials for the details.

---

### Official Review · Reviewer_MLWy · 2022-07-15

**Rating:** 4
**Confidence:** 3
**Soundness:** 2 fair
**Presentation:** 1 poor
**Contribution:** 2 fair

**Summary:**

This paper presents a method for reconstruction a hyperspectral image from its snapshot measurements.

**Questions:**

Introduction should discuss the novelty and technical contributions of the proposed approach over previously proposed approaches.
Theoretical background to support the proposed algorithms should be clearly described.

**Limitations:**

The technical contribution of the proposed approach and theoretical background to support the proposed algorithm are not clear enough from the paper.


**Strengths And Weaknesses:**

This paper requires further improvements on paper organization. The technical contributions of the proposed approach and theoretical background to support the proposed algorithm are not clear enough from the paper. There are many places where the explanation is unclear.

---

> ### Author Response · Authors · 2022-08-02
> **Response to the comments from Reviewer MLWy**
>
> We sincerely thank the reviewer for his/her constructive comments. Please see below for our response.
>
> ***Comments:***
> Introduction should discuss the novelty and technical contributions of the proposed approach over previously proposed approaches. Theoretical background to support the proposed algorithms should be clearly described.
>
> **Responses:**
>
> **Regarding novelty and technical contributions.**  Please refer to the sections of Introduction and Related Work for the discussion on both the novelty and technical contributions of this paper. For convenience, we give a summary here. Overall, we proposed a deep network inspired by unrolled optimization for HSI. The network falls into the category of deep unrolling networks (DUNs), one popular NN architecture in image reconstruction. DNUs are often constructed by unrolling an iterative solver of an optimization problem and replacing some steps with neural network blocks. In our work, based on the unrolled proximal gradient descent (PGD) algorithm, we augment DNUs with two new modules and a new loss for CASSI-based HSI.
> 1.	For the gradient decent steps in PGD, we proposed a ConvLSTM-assistant module, which is inspired by the momentum optimizer. Different from some recent DUNs using LSTM for the proximal mapping or using learnable gradient descent without utilizing cross-stage information, the proposed module can be trained to exploit the information from previous stages to predict the gradient descent step at current stage, which brings noticeable performance improvement.
> 2.	For the proximal mapping in PGD, we proposed a cross-stage self-attention module to improve the feature flow across stages. Different from one latest work [9] using self-attention in an inner-stage manner, the proposed module exploits the query, key, value entries in self-attention with a cross-stage manner, which brings further performance improvement.
> 3.	A spectral geometry consistency loss is proposed, which can regularize the prediction of the DUNs for HSI reconstruction.
>
> **Regarding theoretical background knowledge.** Due to space limitation, the basics of deep unrolling networks and optimization algorithms are just briefly introduced in the paper. We will introduce more background knowledge in supplementary materials in revision.
>
> **While the detailed introduction of the background makes it not very accessible to the readers in this field, it is not essential to the novelty and value of our work. We will make it more accessible in the revision, and hope that the reviewer could reconsider the score based on the responses above and the contributions of our work.**

---

### Official Review · Reviewer_X9GH · 2022-07-17

**Rating:** 6
**Confidence:** 4
**Soundness:** 3 good
**Presentation:** 3 good
**Contribution:** 3 good

**Summary:**

This paper is on spectral reconstruction from snapshot CASSI. It is based on the linear inverse model, and solved by merging model-based and learning-based methods in the framework of deep unrolling network. Compared with existing methods, memory-assistant descent blocks and cross-stage attentive proximal subnetworks are proposed, together with a new loss term. Experiments show that it outperforms exiting unrolling based methods and end-to-end methods.

**Questions:**

Please refer to the weakpoints.

**Limitations:**

No potential negative social impact.

**Strengths And Weaknesses:**

Strong points:
1.	To use deep unrolling for image/video restoration receives attention in recent years, and the authors proposed new add-ons to improve DUN for CASSI based spectral reconstruction.
2.	Experiment results have shown that the proposed method achieves favorable results than existing iterative methods, as well as the latest end-to-end learning based methods.

Weak points:
1.	Initialization of the algorithm and its effect on convergence is not mentioned/discussed.
2.	Model/algorithm complexity and running time is not mentioned, especially the advantages/disadvantages when compared with end-to-end methods.
3.	Many results are directly quoted from other papers. Is there any reason for that? At least ,there are some randomness for data augmentation. To quote results from elsewhere tends to be improper.
4.	No quantitative comparison is conducted on real data. Maybe a real dataset with GT should be prepared first.
5.	Personally, the reviewer has severe concern about the real meaningfulness of trying to reconstruct spectra from CASSI. Given the spectral accuracy in Fig.2 and Fig4, most high-frequency details can not be recovered. To use several mosaiced filters can reach same level accuracy, yet has much better spatial resolution/details. Given this level of spectral details, reconstructed spectral curves are not meaningful for practical applications. One might directly use CASSI raw images for segmentation/recognition directly, without reconstruction.

---

> ### Author Response · Authors · 2022-08-02
> **Response to the comments from Reviewer X9GH**
>
> We sincerely thank the reviewer for his/her constructive comments. Please see below for our point-wise response.
>
> ***1、Initialization of the algorithm and its effect on convergence is not mentioned/discussed.***
>
> In training, the model is initialized by the Kaiming initialization method and $x^{0}$ is initialized by $\phi^\top y$, which is a common practice. The training process converges after 180 epochs. In inference, we also initialize $x^{0}=\phi^\top y$ as it converges faster than using zeros to initialize $x^{0}$. It is empirically observed that after 6 stages, the reconstructed result already achieved SOTA performance, and more stages don't change the result much. We will add such discussions in revision.
>
> ***2、Model / algorithm complexity and running time are not mentioned, especially the advantages / disadvantages when compared with end-to-end methods.***
>
> Please refer to Table 1 in the main paper for the model complexity of different methods in terms of the number of parameters and the number of FLOPS. The observation is that our method has the second smallest number of parameters and the smallest number of FLOPS, among all the compared methods. See the following table for the test time of different methods on the KAIST dataset, which will be added in revision. Clearly, in terms of speed, our model is faster than DNU, DGSMAP and MST-L, and is comparable to TSA-Net. Our model is slower than $\lambda$-Net and HDNet (but with noticeably better performance). The reason is that, currently there is no advanced acceleration support from GPU for the self-attention employed in our model, unlike the standard convolutional layers. This also applies to TSA-Net and MST-L. However, our model has a smaller number of FLOPS than $\lambda$-Net and HDNet, as seen in Table 1 of the main paper. Note that when running on a CPU with a single thread, fewer FLOPs generally lead to shorter inference time. That is why the number of FLOPs is used as the metric in our paper. When running on a GPU, there are many factors in implementation that can influence the inference time, such as parallel programming, sync overhead, memory access cost, number of element-wise operations, and environment support; see the following reference for more details:
> ''Practical Guidelines for Efficient CNN Architecture Design, ECCV 2018.'''
>
> Test time (seconds) and PSNR(dB) of different methods on KAIST dataset.
>
> | Metric | $\lambda$-Net | DNU   | TSA-Net | DNU   | DGSMP | HDNet | MST-L | Ours   |
> |:------:|:------------:|:-----:|:-------:|:-----:|:-----:|:-----:|:-----:|:------:|
> | TIME   | 0.01         | 1.64  | 0.16    | 1.64  | 1.39  | 0.01  | 0.41  | 0.15   |
> | PSNR   | 28.53     | 30.74 | 31.46   | 30.74 | 32.63 | 34.34 | 35.18 | 36.32  |
>
> ***3、Many results are directly quoted from other papers. Is there any reason for that? At least, there are some randomness for data augmentation. To quote results from elsewhere tends to be improper.***
>
> The results of the compared methods are directly quoted only if they are obtained under the same experimental setting as ours. Indeed, many of these results have also been quoted by existing literature when the experimental settings match. We found that those results are reproducible by retraining their models, and the performance difference caused by the randomness of data augmentation is negligible in the experiments.
>
> ***4、No quantitative comparison is conducted on real data. Maybe a real dataset with GT should be prepared first.***
>
> There is no such real-world dataset with GT available in public domain for quantitative evaluation. Building such a dataset also requires a lot of resources and effort. Thus, as with most existing works, we evaluate the performance of our method on real-world measurement data by taking a visual inspection of the results.
>
> ***5、Personally, the reviewer has severe concern about the real meaningfulness of trying to reconstruct spectra from CASSI. Given the spectral accuracy in Fig. 2 and Fig. 4, most high-frequency details cannot be recovered. To use several mosaiced filters can reach same level accuracy, yet has much better spatial resolution/details. Given this level of spectral details, reconstructed spectral curves are not meaningful for practical applications. One might directly use CASSI raw images for segmentation/recognition directly, without reconstruction.***
>
> Thanks for the comment. While the 2D snapshots may provide sufficient information for some applications, hyperspectral images contain additional rich spectral information which benefits many applications such as analyzing the composition of materials. HSI collection/reconstruction with CASSI has the advantages of low cost and fast speed. It is a hot research topic in recent years. We agree that the results of current approaches including ours are not very impressive. However, we expect the research along this line will keep going and the performance of the algorithms will meet practical needs in the future.

---

### Meta-Review · Area_Chair_MccQ · 2022-08-26

**Recommendation:** Reject
**Confidence:** Less certain

**Metareview:**

The paper received mixed reviews with two weak accepts and two borderline rejects, making it a borderline case. Based on the reviews and the rebuttal and on their own reading of the paper, the area chair would like to make a few remarks:
 - the paper achieves good empirical results in terms of PSNR for the task of compressive HSI. This is the main strenght of the paper.
 - the code for reproducing the experiments is not provided. Even though this is not a critical requirement, this would definitely help assessing the quality of the experiments, given that the contribution is mostly methodological.
 - the method consists of modifying several components of classical deep unrolling networks. Some of them are related to existing work, such as the idea to use lstms to exploit past gradients, as discussed in the rebuttal. Such discussions should be included in the paper, even though this idea was not investigated for HSI in this prior work. More generally, the modifications of the DUN method seem quite generic and not specific to compressive HSI. Evaluating their effect on other HSI tasks would be helpful to get a better understanding of the importance of these modifications: are they effective beyond compressive HSI, beyond HSI. If not, why?
 - The rebuttal was useful. Numerous additional experiments were conducted. Yet, it would have been good to include them in a revised version of the pdf.

At this point, it seems that the method is promising, but that a major revision of the paper is required, leading to a reject decision for NeurIPS this year.





**Award:**

No

---

### Decision · Program_Chairs · 2022-09-14

Reject